# Proteomic, Biochemical, and Morphological Analyses of the Effect of Silver Nanoparticles Mixed with Organic and Inorganic Chemicals on Wheat Growth

**DOI:** 10.3390/cells11091579

**Published:** 2022-05-07

**Authors:** Setsuko Komatsu, Hisateru Yamaguchi, Keisuke Hitachi, Kunihiro Tsuchida

**Affiliations:** 1Faculty of Environment and Information Sciences, Fukui University of Technology, Fukui 910-8505, Japan; 2Department of Medical Technology, Yokkaichi Nursing and Medical Care University, Yokkaichi 512-8045, Japan; h-yamaguchi@y-nm.ac.jp; 3Institute for Comprehensive Medical Science, Fujita Health University, Toyoake 470-1192, Japan; hkeisuke@fujita-hu.ac.jp (K.H.); tsuchida@fujita-hu.ac.jp (K.T.)

**Keywords:** proteomics, wheat, silver nanoparticles, chemicals

## Abstract

Wheat is vulnerable to numerous diseases; on the other hand, silver nanoparticles (AgNPs) exhibit a sterilizing action. To understand the combined effects of AgNPs with nicotinate and potassium nitrate (KNO_3_) for plant growth and sterilization, a gel- and label-free proteomics was performed. Root weight was promoted by the treatment of AgNPs mixed with nicotinate and KNO_3_. From a total of 5557 detected proteins, 90 proteins were changed by the mixture of AgNPs, nicotinate, and KNO_3_; among them, 25 and 65 proteins increased and decreased, respectively. The changed proteins were mainly associated with redox and biotic stress in the functional categorization. By immunoblot analysis, the abundance of glutathione reductase/peroxiredoxin and pathogen-related protein three significantly decreased with the mixture. Furthermore, from the changed proteins, the abundance of starch synthase and lipoxygenase significantly increased and decreased, respectively. Through biochemical analysis, the starch contents increased with the mixture. The application of esculetin, which is a lipoxygenase inhibitor, increased the weight and length of the root. These results suggest that the AgNPs mixed with nicotinate and KNO_3_ cause positive effects on wheat seedlings by regulating pathogen-related protein and reactive-oxygen species scavenging. Furthermore, increasing starch and decreasing lipoxygenase might improve wheat growth.

## 1. Introduction

Wheat plays a key role in world-food security since the Green Revolution [1] and the glutens and storage proteins in wheat grain are the source of protein in the human diet [2]. On the other hand, wheat is vulnerable to numerous diseases, such as bacterial and fungal infections, resulting in a considerable drop in yield production and seed quality; additionally, its cultivation is significantly affected by climate change [3]. The wheat-growth period is longer than that of other crop species, including rice and maize, making wheat more susceptible to several diseases and pathogens during its growth and development [4]. The above information indicates that the suppression of pathogens and diseases is important to the increase of wheat growth and development.

The application of AgNPs increased the plant growth parameters as compared to the control, and reduced the percentage of disease incidence as compared to the inoculated control *R. solani* [5]. It was observed that concentrations of 25 and 50 ppm of AgNPs enhanced plant growth and antioxidant levels in mustard [6]. Concentrations of 50 and 75 ppm of AgNPs promoted root nodulation in cowpea and shoot elongation in mustard, respectively [6]. Concentrations of 40 and 60 ppm of AgNPs encouraged root/shoot growth and seed germination in maize, respectively [7]. Concentration of 100 ppm of AgNPs decreased chlorophyll content in tomato, while 1000 ppm increased its superoxide dismutase level [8]. AgNPs lead to the expression of genes related to cell proliferation/metabolism and hormonal signaling in cellular events [9]. AgNPs increased reactive oxygen species’ (ROS) production and total phenolic contents with dose-dependent effects in sugarcane [10]. Because the effect of AgNPs is dependent on their size and concentration, it is crucial to investigate the response to AgNPs on a molecular level to discover their effects on the morphology and physiology of plants.

In plants, organic chemicals such as nicotinate, led to an improvement in the growth and productivity of plants [11]. By regulating fumarase, aconitase, and glutathione metabolism, nicotinate suppressed the oxidative stress in plant cells [12]. In contrast, potassium nitrate (KNO_3_), which is an inorganic chemical, was used for seed priming and dormancy breaking in maize and tomato [13,14]. In most plants, KNO_3_ has an important role in physiological and biochemical processes such as photosynthesis, enzyme activation, energy transfer, and stress resistance [15]. Although the application of organic and inorganic chemicals enhanced plant growth and yield production, their metabolic and molecular mechanism remains unclear.

Currently, it has been reported that the mixture of AgNPs, nicotinate, and KNO_3_ improved soybean growth by regulating the protein-quality control for the misfolded proteins in the endoplasmic reticulum [16]. Additionally, it was reported in wheat leaves that the redox homeostasis was maintained through glycolysis, the activated antioxidant enzymes regulated energy metabolism, and its maintenance of energy-related activities stimulated plant growth in reaction to the mixture of AgNPs, nicotinate, and KNO_3_ [17]. Additionally, the morphological study of next generational wheat plants, which were treated with chemo-blended AgNPs, depicted normal growth, and no toxic effects were observed [17]. Therefore, chemo-blended AgNPs may promote plant growth and development through alteration of plant metabolism. In this study using wheat roots, to elucidate the possible combined effects of AgNPs with nicotinate and KNO_3_, morphological and proteomic analyses were carried out. Furthermore, proteomic results were confirmed with immunoblot and biophysical analyses.

## 2. Materials and Methods

### 2.1. Plant Material and Treatment

Seeds of wheat (*Triticum aestivum* L. cultivar Nourin 61) were sterilized with 2% sodium hypochlorite solution, rinsed with water, and sown in 400 mL of silica sand in a seedling case. Growth conditions were maintained as 16 h light of 200 µmol m^−2^ s^−1^, 8 h dark with 60% humidity at 25 °C. To analyze the effects of AgNPs mixed with organic and inorganic chemicals, AgNPs (US Research Nanomaterials, Houston, TX, USA) with 15 nm, nicotinate (Sigma Aldrich, Darmstadt, Germany), and KNO_3_ (Sigma Aldrich) were used. Groups of 3-day-old plants were treated with a mixture of 5 ppm AgNPs/8 mM nicotinate/0.1 mM KNO_3_, and with the single components, which were 5 ppm AgNPs, 8 mM nicotinate, or 0.1 mM KNO_3_, for 3 days. Untreated plants were used as the control. As morphological parameters, leaf-fresh weight, leaf length, total root-fresh weight, and main-root length were measured at 6 days after sowing (Figure 1). For proteomics, roots were collected (Figure 1). In total, three independent experiments were performed as biological replicates for all experiments. The sowing of seeds was carried out on different days for making biological replicates.

### 2.2. Protein Extraction

A portion (300 mg) of samples was ground in 100 μL of lysis buffer consisting of 7 M urea, 2 M thiourea, 2 mM tributylphosphine, and 5% CHAPS in a filter cartridge plastic rod. The suspension was incubated for 2 min at 25 °C and centrifuged at 15,000× *g* for 5 min at 4 °C twice. Filtrated proteins were used as total-protein extracts. The detergents in protein extracts were removed using the Pierce Detergent Removal Spin Column (Pierce Biotechnology, Rockford, IL, USA). To determine the protein concentration, the Bradford method [18] was used with bovine serum albumin used as the standard.

### 2.3. Protein Enrichment, Reduction, Alkylation, and Digestion

Protein extracts (100 µg) were adjusted to a final volume of 100 µL. After adjustment of protein concentration, proteins were enriched, reduced, alkylated, and digested using the methods described in the previous study (Appendix A) [19]. Peptides were acidified with 1% trifluoroacetic acid and analyzed by nano-liquid chromatography (LC) and mass spectrometry (MS)/MS.

### 2.4. Protein Identification using Nano LC-MS/MS and MS Data Analysis

The peptides were loaded onto the LC system (EASY-nLC 1000; Thermo Fisher Scientific, San Jose, CA, USA) equipped with a trap column (Acclaim PepMap 100 C18 LC column, 3 µm, 75 µm ID × 20 mm; Thermo Fisher Scientific, San Jose, CA, USA) equilibrated with 0.1% formic acid and eluted with a linear acetonitrile gradient (0–35%) in 0.1% formic acid for 120 min at a flow rate of 300 nL/min. The eluted peptides were loaded and separated on the column (EASY-Spray C18 LC column, 3 µm, 75 µm ID × 150 mm; Thermo Fisher Scientific, San Jose, CA, USA) with a spray voltage of 2 kV (Ion Transfer Tube temperature: 275 °C). The peptide ions were detected using MS (Orbitrap Fusion ETD MS; Thermo Fisher Scientific, San Jose, CA, USA) in the data-dependent acquisition mode with the installed Xcalibur software (version 4.0; Thermo Fisher Scientific, San Jose, CA, USA). The protein identification was performed using nano LC-MS/MS with the methods described in the previous study (Appendix A) [20].

The MS/MS searches were carried out using MASCOT (version 2.6.1, Matrix Science, London, UK) and SEQUEST HT search algorithms against *Triticum aestivum* (SwissProt TaxID = 4565_and_subtaxonomies) (version 2017-07-05) protein database, the size of which is SwissProt = 370, TrEBML = 145,221, and total = 145,591, using Proteome Discoverer (PD) 2.2 (version 2.2.0.388; Thermo Scientific, San Jose, CA, USA). The condition of the analysis is described in the previous study (Appendix A) [20].

### 2.5. Differential Analysis of Proteins Using MS Data

Label-free quantification was performed with PD 2.2 using precursor ions’ quantifier nodes. For differential analysis of the relative abundance of peptides and proteins between samples, the free available software Perseus (version 1.6.0.7; Max Planck Institute of Biochemistry, Martinsried, Germany) [21] was employed. The condition of analysis is described in the previous study (Appendix A) [20]. Principal component analysis (PCA) was performed with Perseus. The gene functional annotation and protein categorization were analyzed using MapMan bin codes [22].

### 2.6. Immunoblot Analysis

An SDS-sample buffer consisting of 60 mM Tris-HCl (pH 6.8), 2% SDS, 10% glycerol, 5% dithiothreitol, and bromophenol blue was added to the protein extract [23]. Protein extracts (10 µg) were separated on a 12% SDS-polyacrylamide gel electrophoresis (PAGE) and transferred onto a polyvinylidene-difluoride membrane. The membrane was blocked in Bullet Blocking One reagent (Nacalai Tesque, Kyoto, Japan) for 5 min. After blocking, the membrane was cross-reacted with the primary antibodies for 30 min. As primary antibodies, anti-glutathione reductase (Agrisera, Vännäs, Sweden), anti-ascorbate peroxidases [24], anti-peroxiredoxin [25], anti-pathogenesis related protein 1 (PR1 protein) [26], anti-PR3 protein [26], and anti-PR5 protein antibodies [26] were used. As secondary antibody, anti-rabbit IgG conjugated with horseradish peroxidase (Bio-Rad, Hercules, CA, USA) was used. After reaction for 30 min, the membrane was reacted with the TMB Membrane Peroxidase Substrate kit (SeraCare, Milford, MA, USA) for 1–10 min. The integrated densities of bands were calculated using ImageJ software (version 1,53e/Java 1.8.0 172; National Institutes of Health, Bethesda, MD, USA). The pattern of Coomassie brilliant blue staining was used as a loading control.

### 2.7. Starch-Content Assay and Starch Staining

A portion (10 mg) of leaves was homogenized in phosphate-buffered saline with a mortar and pestle. Starch content was analyzed using a Starch Assay Kit (BioAssay Systems, Hayward, CA, USA). After washing off free glucose and small oligosaccharides with 1 mL of 90% ethanol, homogenates were warmed to 60 °C for 5 min with occasional vortexing. After centrifugation at 10,000× *g* for 2 min, soluble starch in the pellet was extracted with 1 mL of water incubated in a boiling water bath for 5 min and spun at 10,000× *g* for 2 min. The supernatant was collected as soluble starch. After extracting soluble starch, the water insoluble pellet was extracted with 0.2 mL of dimethyl sulfoxide and heated in boiling water bath for 5 min. Alternatively, resistant starch was extracted with KOH and acetate. Each 10 μL sample was added to a working reagent by mixing 90 μL of assay buffer, 1 μL of enzyme A, 1 μL of enzyme B, and 1 μL of dye reagent. The mixture was incubated for 30 min at room temperature and the absorbance of the mixture was measured at 570 nm.

For the starch staining of the leaves, they were wormed in 80 °C ethanol for 10 min and washed with water. After softening the leaves, they were stained with 0.01 N iodine solution. The starch-accumulation ratio was calculated with the ratio (%) of shoot length with changed color/shoot length.

### 2.8. Application of Lipoxygenase Inhibitor

As a lipoxygenase inhibitor, esculetin (Tokyo Chemical, Tokyo, Japan) was used. After sowing, 3-day-old plants were treated with or without 0.56, 5.6, and 56 µM esculetin for 3 days. As morphological parameters, leaf-fresh weight, leaf length, total root-fresh weight, and main-root length were measured at 6 days after sowing. Untreated plants were used as the control.

### 2.9. Statistical Analysis

Data were analyzed by one-way ANOVA followed by Tukey’s multiple comparisons among multiple groups using SPSS (version 22.0; IBM, Armonk, NY, USA). A *p*-value of less than 0.05 was considered as statistically significant.

## 3. Results

### 3.1. Growth Response of Wheat to AgNPs Mixed with Organic and Inorganic Chemicals

To evaluate the effects of AgNPs mixed with organic and inorganic chemicals on wheat, a morphological analysis was conducted (Figure 1). Three-day-old plants were treated with or without 5 ppm AgNPs, 8 mM nicotinate, 0.1 mM KNO_3_, and 5 ppm AgNPs/8 mM nicotinate/0.1 mM KNO_3_ for 3 days. Leaf-fresh weight, leaf length, total root-fresh weight, and main-root length were measured at 6 days after sowing (Figure 2). Leaf-fresh weight, leaf length, and main-root length were enhanced by nicotinate, KNO_3_, AgNPs, or the mixture of AgNPs/nicotinate/KNO_3_. Furthermore, total root-fresh weight was significantly enhanced by AgNPs mixed with nicotinate and KNO_3_ compared with untreated plant and other single treatments (Figure 2). Based on this result, roots were used for the following proteomic analysis.

### 3.2. Protein Responses of Wheat Roots to AgNPs Mixed with Organic and Inorganic Chemicals

To understand the mechanism caused by AgNPs mixed with nicotinate and KNO_3_ on wheat, a gel- and label-free proteomic analysis was performed (Appendix A). By proteomic analysis, 5557 proteins were identified in wheat roots (Appendix A). The proteomic results of six samples from control and treated groups were compared by PCA, which indicated different accumulation patterns of proteins between the control and treatment (Appendix A). In the differential analysis, 90 proteins were significantly changed by AgNPs mixed with nicotinate and KNO_3_ compared with the control. Among 90 proteins, 25 and 65 proteins increased and decreased, respectively, by AgNPs mixed with nicotinate and KNO_3_ compared with the control (Appendix A).

The identified proteins were functionally categorized using MapMan bin codes (Appendix A, Figure 3). The differentially changed proteins were mainly detected in redox and biotic stress in the functional category (Figure 3). From the changed proteins, the abundance of starch synthase and lipoxygenase significantly increased and decreased, respectively (Appendix A). Based on these results, proteins categorized in redox and biotic stress were further confirmed using immunoblot analysis. Additionally, the roles of starch synthase and lipoxygenase, which were significantly changed by AgNPs mixed with nicotinate and KNO_3_, were analyzed using biophysical analysis.

### 3.3. Amount of ROS Scavenging Enzymes in Wheat Treated with AgNPs Mixed with Organic and Inorganic Chemicals

To confirm the changes in the amount of ROS scavenging enzymes in wheat treated with or without AgNPs/nicotinate/KNO_3_, immunoblot analysis was performed. Because glutathione peroxidase and catalase were identified as increased proteins by proteomic analysis, other proteins, which were ascorbate peroxidase, glutathione reductase, and peroxiredoxin were analyzed by immunoblot analysis. Three-day-old wheats were treated with or without AgNPs/nicotinate/KNO_3_, and proteins were extracted from leaves and roots at 3 days of treatment. Protein extracts were separated on SDS-PAGE, transferred onto membranes, and cross-reacted with anti-ascorbate peroxidase, anti-glutathione reductase, and anti-peroxiredoxin antibodies (Appendix A). The relative band intensities were calculated (Figure 4). The amount of 50-kDa glutathione reductase significantly decreased in leaves with the treatment of AgNPs mixed with nicotinate and KNO_3_ compared with other treatments and the control. The amount of 25-kDa peroxiredoxin also decreased in roots with the treatment of only AgNPs or AgNPs mixed with nicotinate and KNO_3_ compared with the control. On the other hand, 25-kDa ascorbate peroxidase increased in the root by AgNPs and it did not change with the treatment of AgNPs mixed with nicotinate and KNO_3_ (Figure 4).

### 3.4. Accumulation of PR Proteins in Wheat Treated with AgNPs Mixed with Organic and Inorganic Chemicals

To confirm the changes in the accumulation pattern of PR proteins in wheat treated with or without AgNPs/nicotinate/KNO_3_, immunoblot analysis was performed. Because chitinase named PR3 and thaumatin named PR5 were identified as decreased proteins by proteomic analysis, these two proteins as well as PR1 were analyzed by immunoblot analysis. Three-day-old wheats were treated with or without AgNPs/nicotinate/KNO_3_, and proteins were extracted from leaves and roots at 3 days of treatment. Protein extracts were separated on SDS-PAGE, transferred onto membranes, and cross-reacted with anti-PR1 protein, anti-PR3 protein, and anti-PR5 protein antibodies (Appendix A). The relative band intensities were calculated (Figure 5). The abundance of PR3 significantly decreased in roots with the treatment of AgNPs mixed with nicotinate and KNO_3_ compared with other treatments and the control (Figure 5). However, other PR proteins did not change in leaves and roots with the treatment of AgNPs mixed with nicotinate and KNO_3_.

### 3.5. Starch Accumulation and Contents in Wheat Leaves Treated with AgNPs Mixed with Organic and Inorganic Chemicals

To analyze the change of starch synthetase, whose abundance was significantly increased by the treatment of AgNPs mixed with nicotinate and KNO_3_ compared with untreated plant (Appendix A), the accumulation and contents of starch were measured in leaves (Figure 6). Three-day-old wheats were treated with or without AgNPs/nicotinate/KNO_3_ for 3 days. Leaves were stained with iodine solution and starch contents were measured (Figure 6). Through biochemical analyses, the starch accumulation and starch contents significantly increased with the treatment of AgNP mixtures with nicotinate and KNO_3_ compared with other treatments and the control (Figure 6).

### 3.6. The Effect of Lipoxygenase Inhibitor on Growth of Wheat Treated with AgNPs Mixed with Organic and Inorganic Chemicals

Out of the differentially changed proteins, the abundance of lipoxygenase significantly decreased (Appendix A). To analyze the role of lipoxygenase on wheat growth, esculetin, which is lipoxygenase inhibitor, was applied (Figure 7). Three-day-old plants were treated with or without 0.56, 5.6, and 56 µM esculetin for 3 days. Leaf-fresh weight, leaf length, total root-fresh weight, and main-root length were measured. The application of 0.56 µM esculetin increased total root-fresh weight and main-root length (Figure 7). However, leaf-fresh weight and leaf length did not change by the application of esculetin.

## 4. Discussion

### 4.1. AgNPs Mixed with Organic and Inorganic Chemicals Enhance the Root Growth of Wheat

The AgNPs have a wide range of applications as an essential component in many different products, such as household, food, and industrial goods, because of their fungicidal and bactericidal properties [27,28]. Compared to Ag-based compounds, AgNPs with increased surface area available for microbe interaction are more toxic to bacteria, fungi, and viruses. Furthermore, AgNPs exhibit sterilizing action. To illustrate the effects of AgNPs mixed with organic and inorganic chemicals on wheat growth, morphological analysis was performed. Among wheat growth parameters, root growth was promoted in response to AgNPs mixed with organic and inorganic chemicals (Figure 2).

The effects of AgNPs on plants largely depend on the plant species, plant-growth stages, and composition/concentration of the NPs [29]. Although AgNPs’ exposure was detrimental for plant growth, the growth-promoting properties of AgNPs were demonstrated in *Eruca sativa* [29], mustard [30], wetland plants [31], maize and *Phaseolus vulgaris* [32], and soybean [33,34,35]. Application of 5 ppm AgNPs mixed with organic and inorganic chemicals enhanced the growth of soybean [16] and wheat [17]. The current results (Figure 2) with previous studies suggest that AgNPs promote wheat growth; furthermore, additional nicotinate and KNO_3_ can enhance this growth, especially on wheat root.

### 4.2. ROS Scavenging Pathway Is Suppressed in Wheat Treated with AgNPs Mixed with Organic and Inorganic Chemicals

AgNPs induced oxidative stress in bacteria, animals, algae, and plants [36]. AgNPs increased ROS production and total phenolic contents with dose-dependent effects in sugarcane [10]. In tomatoes, 100 ppm AgNPs decreased chlorophyll contents, while 1000 ppm increased the super oxide dismutase activity [8]. In this study, ascorbate peroxidase (Figure 4) increased in wheat root with AgNPs and it did not change with the treatment of AgNPs mixed with nicotinate and KNO_3_ compared with the control. The application of AgNPs generated ROS in the cell, leading to both positive and negative effects on plant growth. However, factors such as size, shape, surface coating, and concentration of AgNPs as well as plant species vary among studies resulting in conflicting reports of the effect at times.

The induction of ROS by AgNPs is mainly produced in the chloroplast, followed by a reduction of the solar-energy consumption, after which the excess-excitation energy promotes ROS production in the chloroplast [37]. On the other hand, the activities of antioxidant enzymes in AgNPs- and biochar-treated plants suppressed the oxidative stress [38]. Biochar surface carboxyl and sulfur functional groups were involved in a complexation process with AgNPs, which inhibited the oxidative dissolution and Ag-ions release alongside the biochar space shield effect [38]. Furthermore, phytochemicals-capped AgNPs acted as a growth promoter at lower concentrations by delivering a potent antioxidant during early stage of seedling as compared to chemically synthesized AgNPs-treated seedlings of wheat [39]. The current results with previous reports suggest that ROS induced by AgNPs might be suppressed by ascorbate peroxidase in wheat root with the addition of nicotinate and KNO_3_. Furthermore, amounts of glutathione reductase and peroxiredoxin decreased in the leaf and root, respectively, which might be a result of ROS-stress reduction by the application of the AgNPs mixture with nicotinate and KNO_3_.

### 4.3. PR Protein Is Reduced in Wheat Treated by AgNPs Mixed with Organic and Inorganic Chemicals

PR proteins are an integral part of the defense mechanisms of plants against various types of abiotic and biotic stresses. Plants evolved different kinds of defense mechanisms, including physical and chemical defenses, to protect themselves from pathogens, which terminate pathogen infection and disease development [40]. Plant-pathogen interactions result in the activation of defense-signaling pathways such as salicylate and jasmonate signaling pathways. This activation leads to the accumulation of PR proteins, which inhibit the growth of pathogens or repress the spread of disease to other organs [41]. The *PR* genes such as *PR1*, *PR3*, *PR5*, and *PR12* are signature genes of the salicylate and jasmonate signaling pathways. The salicylate-dependent signaling pathway controls the systemic acquired resistance (SAR) via the induction of the expression of salicylate-response *PR* genes such as *PR1*, *PR4*, and *PR5*, which are associated with wide-spectrum plant-defense responses [42]. On the other hand, the jasmonate/ethylene-dependent signaling pathway regulates the induced systemic resistance (ISR), which is associated with the expression of jasmonate/ethylene-response genes such as *PR3*, *PR4*, and *PR12* [43]. In this study, PR3 protein significantly decreased in roots with the treatment of AgNPs mixed with nicotinate and KNO_3_ compared with other treatments and the control (Figure 5). These results with previous reports suggest that the jasmonate/ethylene-dependent signaling pathway is related to AgNPs signaling and it is suppressed in wheat root with the addition of nicotinate and KNO_3_.

PR2 and PR3 genes encoded β-1,3-glucanase [44] and chitinase [45], respectively. Molecular docking analysis of β-1,3-glucanase and chitinase with beta-glucan and chitin revealed crucial amino acid residues involved in ligand binding and important interactions, indicating their key role in wheat defense against fungal [46]. Β-1,3-glucanase and chitinase are the best soldiers against fungal. In addition, β-1,3-glucanase and chitinase produced by bacteria [47] and fungi [48] also induced high antifungal activities in plants. In this study, the abundance of chitinase named PR3 and thaumatin named PR5 were suppressed by AgNPs mixed with nicotinate and KNO_3_ (Appendix A, Figure 5). These results with previous findings suggest that wheat can grow as the result of the defense against the fungal pathogen by the application of the AgNPs mixture with nicotinate and KNO_3_. They also suggest that the mixture treatment with AgNPs, nicotinate, and KNO_3_ might be a useful combination for wheat defense against fungal pathogens.

### 4.4. AgNPs Mixed with Organic and Inorganic Chemicals Enhance the Accumulation of Starch in Wheat Leaves

Plants achieve tolerance to an adverse environment through energy metabolism. Starch, which is the main energy storage substance, widely exists in most plants to regulate their growth/development and increase stress tolerance. In broad-bean seedlings, AgNPs adversely affected the chloroplast ultrastructure, but increased plant growth and starch accumulation [49]. In this study, starch accumulated in the leaves of wheat treated with AgNPs mixed with organic and inorganic chemicals (Figure 6). The effects of AgNPs on the seed-germination stage led to the creation of nanopores for enhanced water uptake, the rebooting of ROS/antioxidant systems, the generation of hydroxyl radicals for cell wall loosening, and nanocatalyst for fastening starch hydrolysis [50]. However, after germination, when 3-day-old wheat was treated with AgNPs, the plant growth mechanism might be different with the ROS-scavenging condition and sucrose accumulation. These findings suggest that AgNPs mixed with organic and inorganic chemicals alter the sugar metabolism of wheat, which might promote the accumulation of soluble sugars and positively facilitate wheat growth through the upregulation of glycolysis.

### 4.5. AgNPs Mixed with Organic and Inorganic Chemicals Enhance Wheat-Root Growth through Lipoxygenase Regulation

Several coumarins and flavonoids acted as lipoxygenase inhibitors; in particular, esculetin and quercetin were potent inhibitors [51]. In this study, the abundance of lipoxygenase decreased with AgNPs mixed with organic and inorganic chemicals; and the application of esculetin increased root weight and length without AgNPs (Figure 7). In plants, lipoxygenases catalyze dioxygenation of linoleic (C18:2) and linolenic (C18:3) acids to form the corresponding hydroperoxy fatty acids, which are used for oxylipin synthesis [52]. Jasmonates, which are one of the oxylipins, are derived from 13-hydroperoxides through the plastidial type-2 lipoxygenases pathway [53]. Lipoxygenases function similarly to vegetative storage proteins and perform a crucial role during the maturation of seeds and seedling growth [54]. During the early stage of seedling growth in several crops, various novel lipoxygenases are induced [55]. Because linoleic and linolenic acid act as lipoxygenase substrates [55], lipoxygenase might induce plant growth in wheat by the application of AgNPs with mixed with organic and inorganic chemicals. These results with previous findings suggest that the wheat-root growth promoted by AgNPs mixed with organic and inorganic chemicals might be regulated through jasmonates derived from the lipoxygenases pathway.

## 5. Conclusions

Wheat is vulnerable to numerous diseases; on the other hand, AgNPs exhibit sterilizing action. To understand the effects of AgNPs mixed with nicotinate and KNO_3_ for plant growth and sterilization, morphology was measured. Root weight was enhanced by treatment with a mixture of AgNPs, nicotinate, and KNO_3_. According to proteomics with the mixture, changed proteins were functionally categorized with redox and biotic stress as increased and decreased proteins, respectively. Furthermore, the abundance of starch synthase and lipoxygenase significantly increased and decreased, respectively. Based on the proteomic results, confirmation experiments were biochemically performed in wheat with the application of AgNPs, nicotinate, and KNO_3_. The principal findings of this research are as follows: (i) accumulation of glutathione reductase/peroxiredoxin decreased, while catalase and ascorbate peroxidase increased; (ii) accumulation of PR2 and PR3 protein decreased; (iii) starch contents increased in the leaves; and (iv) the application of a lipoxygenase inhibitor increased root weight and length. Current results indicate that the mixture of AgNPs, nicotinate, and KNO_3_ enhances the growth of wheat seedlings by regulating the PR protein and ROS scavenging system. Furthermore, increasing starch and decreasing lipoxygenase might improve wheat growth.

## Figures and Tables

**Figure 1 cells-11-01579-f001:**
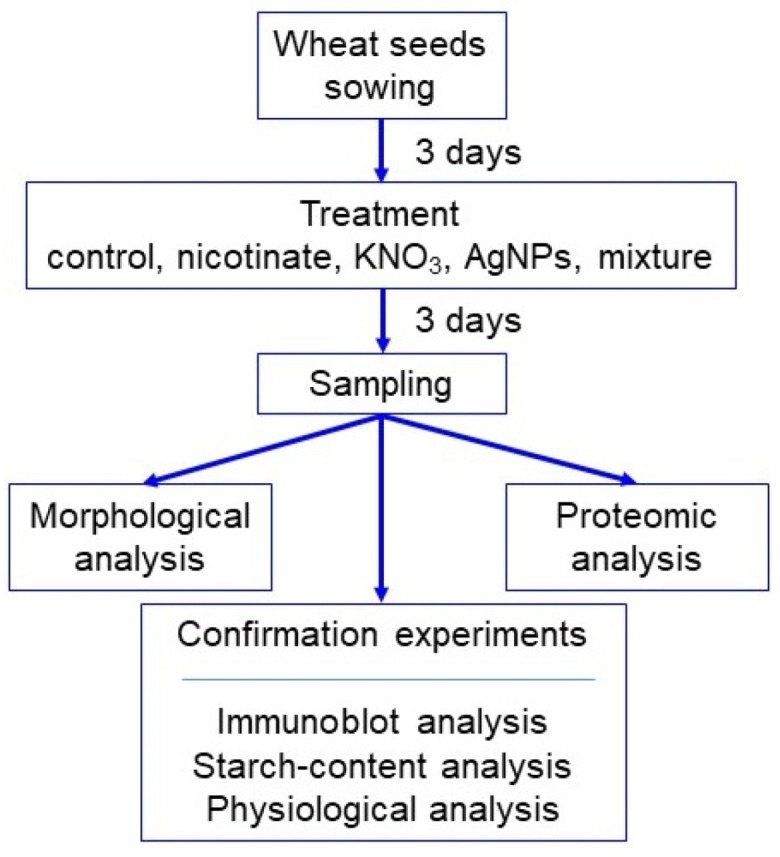
Experimental design for morphological, proteomic, and confirmation analyses of wheat treated with the mixture of AgNPs, nicotinate, and KNO_3_. Three-day-old wheat seedlings were treated with and without 8 mM nicotinate, 0.1 mM KNO_3_, 5 ppm AgNPs, and 5 ppm AgNPs/8 mM nicotinate/0.1 mM KNO_3_ (mixture) for 3 days. Wheat seedlings were analyzed with morphology and proteomics. Proteomic results were confirmed by immunoblot, starch-content, and morphophysiological analyses. All experiments were performed with 3 independent biological replicates.

**Figure 2 cells-11-01579-f002:**
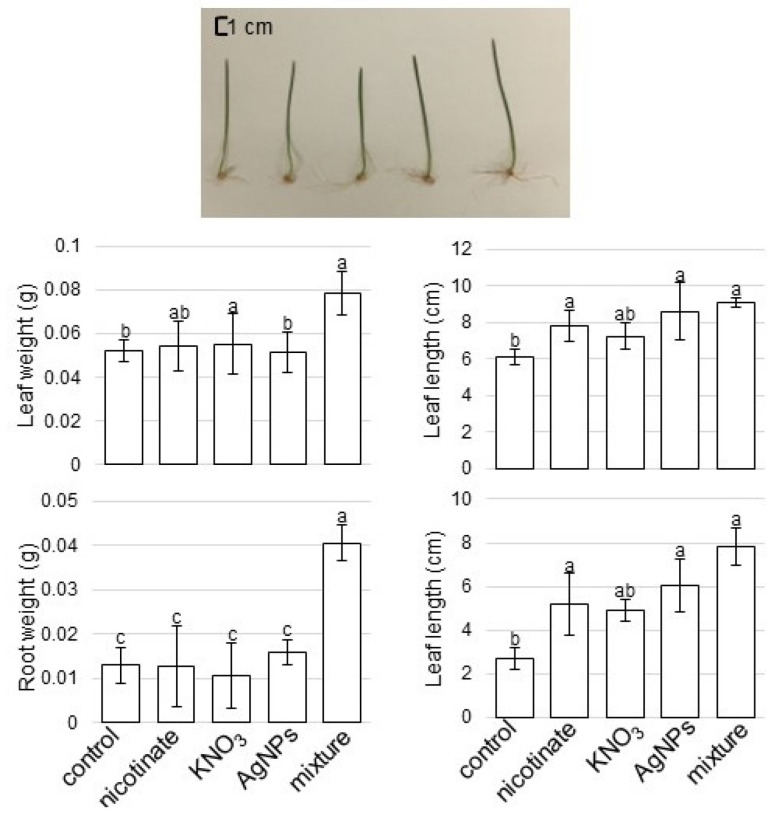
Morphological change of wheat treated with AgNPs, nicotinate, and KNO_3_. Three-day-old wheat seedlings were treated with and without 8 mM nicotinate, 0.1 mM KNO_3_, 5 ppm AgNPs, and 5 ppm AgNPs/8 mM nicotinate/0.1 mM KNO_3_ (mixture) for 3 days. Leaf-fresh weight, leaf length, total root-fresh weight, and main-root length were measured. Scale bar in the picture shows 1 cm. The data are presented as mean ± SD from 3 independent biological replicates. Mean values of each point with different letters are significantly different according to one-way ANOVA followed by Tukey’s multiple comparison test (*p* < 0.05).

**Figure 3 cells-11-01579-f003:**
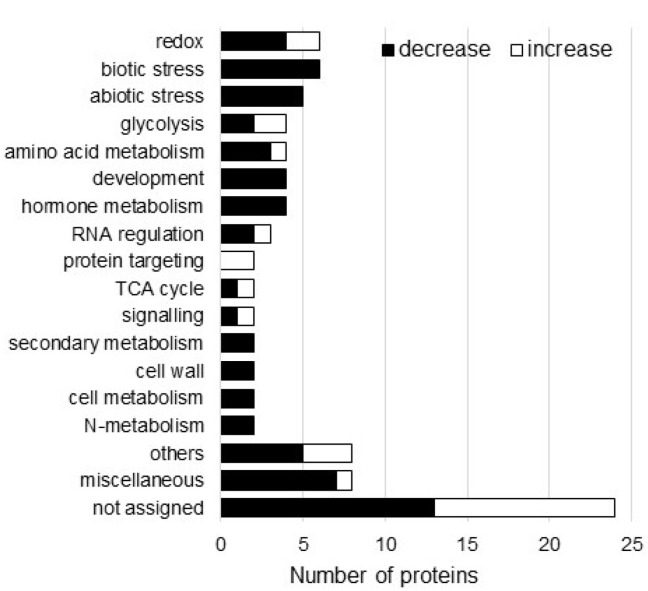
Functional categories of proteins with differential abundance in wheat root treated with AgNPs/nicotinate/KNO_3_. Three-day-old wheat seedlings were treated with and without the mixture of 5 ppm AgNPs, 8 mM nicotinate, and 0.1 mM KNO_3_ for 3 days. Proteins extracted from the root were analyzed using a gel-free/label-free proteomic technique. The significantly changed proteins were identified (*p* < 0.05) and functionally categorized using MapMan bin codes (Appendix A). The x-axis indicates the number of identified proteins. White and black columns mean the number of increased and decreased proteins, respectively, proteins by treatment with 5 ppm AgNPs/8 mM nicotinate/0.1 mM KNO_3_ compared with untreated plant. Abbreviation: TCA cycle, tricarboxylic acid cycle.

**Figure 4 cells-11-01579-f004:**
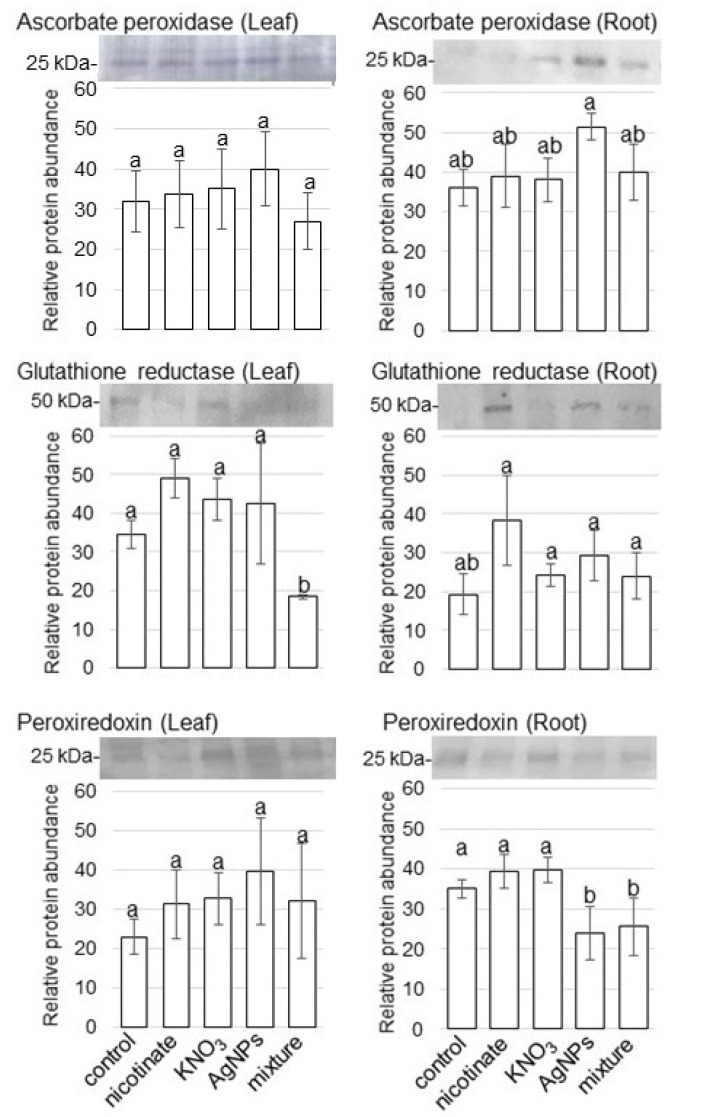
Accumulation of ROS scavenging enzymes in wheat treated with AgNPs/nicotinate/KNO_3_. Three-day-old wheat seedlings were treated with and without 8 mM nicotinate, 0.1 mM KNO_3_, 5 ppm AgNPs, and 5 ppm AgNPs/8 mM nicotinate/0.1 mM KNO_3_ (mixture) for 3 days. Proteins (10 µg) extracted from leaves and roots were separated on SDS-PAGE and immunoblot analyses were performed with anti-ascorbate peroxidase, glutathione reductase, and peroxiredoxin antibodies. The integrated densities of bands were calculated using ImageJ software. The data are presented as mean ± SD from 3 independent biological replicates (Appendix A). Statistical analysis is the same as Figure 2.

**Figure 5 cells-11-01579-f005:**
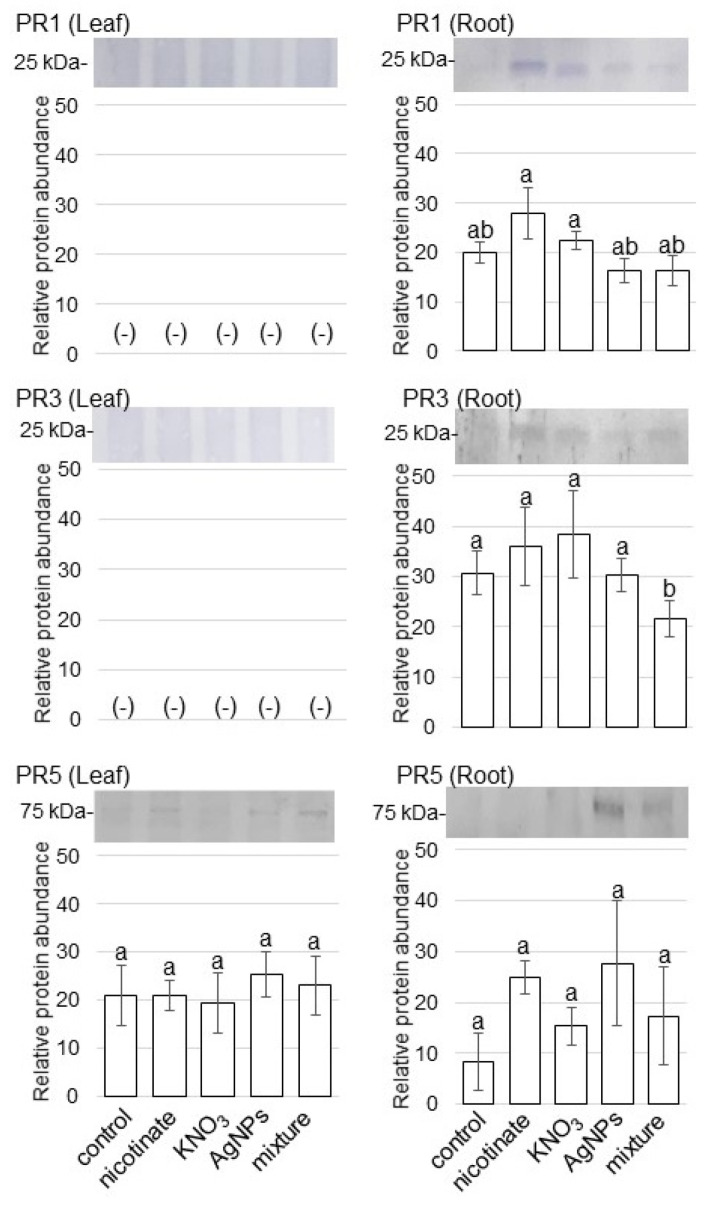
Accumulation of PR proteins in wheat treated with AgNPs/nicotinate/KNO_3_. Three-day-old wheat seedlings were treated with and without 8 mM nicotinate, 0.1 mM KNO_3_, 5 ppm AgNPs, and 5 ppm AgNPs/8 mM nicotinate/0.1 mM KNO_3_ (mixture) for 3 days. Proteins (10 µg) extracted from leaves and roots were separated on SDS-PAGE and immunoblot analyses were performed with anti-PR1, PR3, and PR5 antibodies. The integrated densities of bands were calculated using Image J software. The data are presented as mean ± SD from 3 independent biological replicates (Appendix A). Statistical analysis is the same as Figure 2.

**Figure 6 cells-11-01579-f006:**
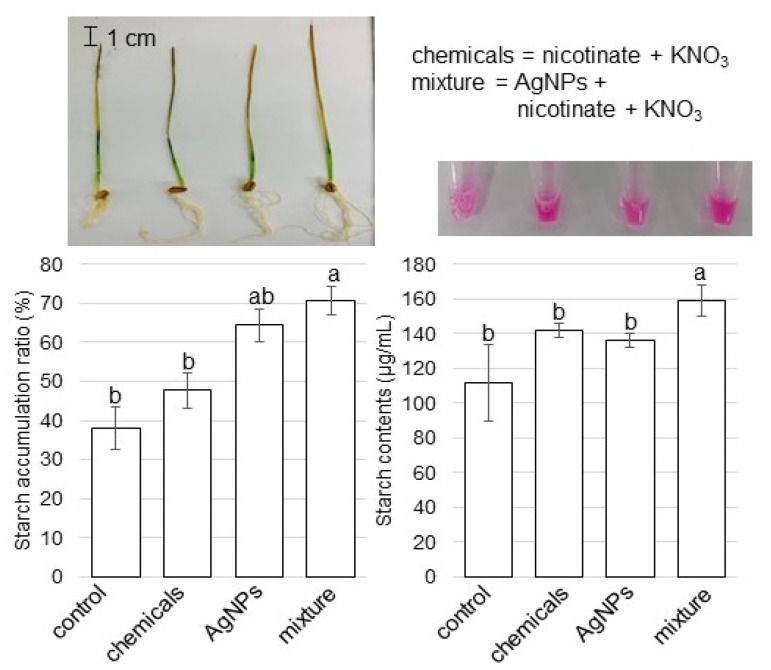
Accumulation ratio and contents of starch in leaves of wheat treated with AgNPs/nicotinate/KNO_3_. Three-day-old wheat seedlings were treated with and without 8 mM nicotinate/0.1 mM KNO_3_ (chemicals), 5 ppm AgNPs, and 5 ppm AgNPs/8 mM nicotinate/0.1 mM KNO_3_ (mixture) for 3 days. Starch content was analyzed using a Starch Assay Kit and starch accumulation was measured with leaves stained by iodine solution. Scale bar in the picture shows 1 cm. The data are presented as mean ± SD from 3 independent biological replicates. Statistical analysis is the same as Figure 2.

**Figure 7 cells-11-01579-f007:**
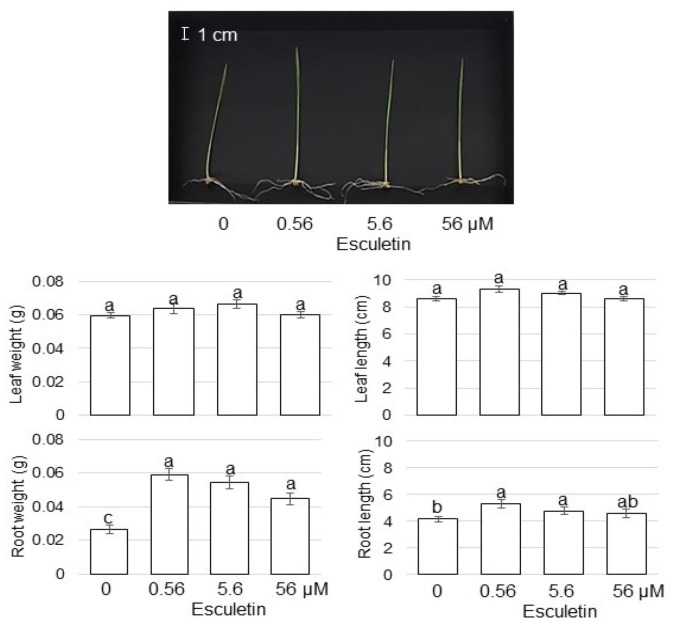
Morphological change by application of lipoxygenase inhibitor to wheat. As lipoxygenase inhibitor, esculetin, was used. Three-day-old wheat seedlings were treated with or without 0.56, 5.6, and 56 µM escletin for 3 days. Leaf-fresh weight, leaf length, total root-fresh weight, and main-root length were measured. Scale bar in the picture shows 1 cm. The data are presented as mean ± SD from 3 independent biological replicates. Statistical analysis is the same as Figure 2.

## Data Availability

MS data, RAW data, peak lists, and result files have been deposited in the ProteomeXchange Consortium [56] via the jPOST [57] partner repository under data-set identifiers PXD008949.

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
