# Peer review of "Proteomic, Biochemical, and Morphological Analyses of the Effect of Silver Nanoparticles Mixed with Organic and Inorganic Chemicals on Wheat Growth"

_cells, 2022, doi:10.3390/cells11091579_

Round 1

Reviewer 1 Report

The manuscript is clear and well written, but the authors should mark the major innovations reported in this work compared to the results already present in the literature.

The manuscript "Proteomic, Biochemical, and Morphological Analyses of the Ef-2 fect of Silver Nanoparticles Mixed with Organic and Inorganic 3 Chemicals on Wheat Growth" by Komatsu at al. reports the results of a series of tests performed on wheat plants after treatment with AgNPs mixed with nicotinate and KNO3 with respect to control samples and the treatment with the single substances (AgNPs, nicotinate and KNO3). In particular, they observed that the mixture of AgNPs with nicotinate and KNO3 have positive effects on the wheat growth seedlings and starch contents by regulating the PR protein and ROS scavenging system. 

The manuscript reports interesting results, but it looks like a simple scientific report of the obtained results after some standard tests. I think that the discussion should be extended and the authors should provide an explanation of the results. What are the synergic effects of AgNPs, nicotinate and KNO3 mixture on the overall growth of wheat plants, starch contents, ROS and PR protein accumulation; what are the mechanisms involved in these processes? The authors should clearly explain what is the major novelty of this work with respect to others reported in the literature.

Some minor typos should be corrected and some part of the text should be discussed more in details for who is not familiar with the used terminologies:

-Page 2 rows 43-44: what you mean for AgNPS with 25 and 50 ppm (50 and 75ppm)? Do you mean the concentration of AgNPS? I think that it is better writing “It was observed that concentrations of 25 and 50 ppm of AgNPs enhanced….

-Page 2 rows76-77: you should explain clearly that the plants were treated with the mixture and with the single components (AgNPs, nicotinate and KNO3). This is not immediately clear from this first sentence.

-Figures 2,4,5,6,7 what is the meaning of letters a, b reported above the bars of the graphics?

-Figure S1 is not clear, can you provide an image with higher resolution? Can you explain the graph of the picture? I don’t understand the meaning of the blue and red curves.

-Page 4 what do you mean when you write that among 90 proteins, 25 and 65 increased and decreased, respectively? What is the parameter that increases or decreases? The number of proteins?

-Page 8: the first paragraph of the discussion is a repetition of the introduction, please change it or remove it.

-Provide the meaning of PCA and ISR

-Page 10 row 275 correct “previous” with “previous”

Author Response

Reviewer 1

The manuscript is clear and well written, but the authors should mark the major innovations reported in this work compared to the results already present in the literature.

The manuscript "Proteomic, Biochemical, and Morphological Analyses of the Ef-2 fect of Silver Nanoparticles Mixed with Organic and Inorganic 3 Chemicals on Wheat Growth" by Komatsu at al. reports the results of a series of tests performed on wheat plants after treatment with AgNPs mixed with nicotinate and KNO3 with respect to control samples and the treatment with the single substances (AgNPs, nicotinate and KNO3). In particular, they observed that the mixture of AgNPs with nicotinate and KNO3 have positive effects on the wheat growth seedlings and starch contents by regulating the PR protein and ROS scavenging system. 

The manuscript reports interesting results, but it looks like a simple scientific report of the obtained results after some standard tests. I think that the discussion should be extended and the authors should provide an explanation of the results.

What are the synergic effects of AgNPs, nicotinate and KNO3 mixture on the overall growth of wheat plants, starch contents, ROS and PR protein accumulation;

what are the mechanisms involved in these processes?

Answer: Thank you very much for your critical comments. The discussion sections, “4.2” for ROS scavenging, “4.3” for PR proteins, and “4.4” for starch contents, have been extended as suggested. The corrected parts have been marked in the text in red.

The authors should clearly explain what is the major novelty of this work with respect to others reported in the literature.

Answer: Thank you very much for additional critical comments. Because it is never reported in previous publication that AgNPs mixed with organic and inorganic chemicals enhance wheat-root growth through lipoxygenase regulation, this finding is the major novelty. The discussion section “4.5” has been re-written and marked in red.  

Some minor typos should be corrected and some part of the text should be discussed more in details for who is not familiar with the used terminologies:

Answer: I am sorry that we made mistakes in the text. They have been corrected carefully. Additionally, the discussion section has been examined in detail. The corrected parts have been marked in the text in red.

-Page 2 rows 43-44: what you mean for AgNPS with 25 and 50 ppm (50 and 75ppm)? Do you mean the concentration of AgNPS? I think that it is better writing “It was observed that concentrations of 25 and 50 ppm of AgNPs enhanced….

Answer: Thank you very much for your suggestion. They have been corrected as suggested: “It was observed that concentrations of 25 and 50 ppm of AgNPs enhanced plant growth and antioxidant levels in mustard [6]. Concentrations of 50 and 75 ppm of AgNPs promoted root nodulation in cowpea and shoot elongation in mustard, respectively [6]. Concentrations of 40 and 60 ppm of AgNPs encouraged root/ shoot growth and seed germination in maize, respectively [7]. Concentration of 100 ppm of AgNPs decreased chlorophyll content in tomato, while 1000 ppm increased its superoxide dismutase level [8].”

-Page 2 rows76-77: you should explain clearly that the plants were treated with the mixture and with the single components (AgNPs, nicotinate and KNO3). This is not immediately clear from this first sentence.

Answer: As suggested, they have been corrected as follows: “Groups of 3-day-old plants were treated with a mixture of 5 ppm AgNPs/ 8 mM nicotinate/ 0.1 mM KNO3, and with the single components, which were 5 ppm Ag NPs, 8 mM nicotinate, or 0.1 mM KNO3, for 3 days. Untreated plants were used as control.”

-Figures 2,4,5,6,7 what is the meaning of letters a, b reported above the bars of the graphics?

Answer: As suggested, the explanation of alphabet letters has been added in the legends of Figure 2. The meaning is as follows: “Mean values of each point with different letters are significantly different according to one-way ANOVA followed by Tukey’s multiple comparison test (p<0.05).” Additionally, the following sentence has been added in Figures 4, 5, 6, and 7: “Statistical analysis is the same as Figure 2.”

-Figure S1 is not clear, can you provide an image with higher resolution? Can you explain the graph of the picture? I don’t understand the meaning of the blue and red curves.

Answer: We are sorry that Figure S1was not clear and not explained. Because the Venn diagram was automatically generated by the software, an image with high resolution was not prepared; so this graph has been re-written by ourselves. Additionally, the explanation of the PCA picture has been added. The blue color and orange color show control group and mixture group, respectively.     

-Page 4 what do you mean when you write that among 90 proteins, 25 and 65 increased and decreased, respectively? What is the parameter that increases or decreases? The number of proteins?

Answer: Thank you very much for your highlighting this problem. These sentences have been corrected as follows: “In the differential analysis, 90 proteins were significantly changed by AgNPs mixed with nicotinate and KNO3 compared with control. Among 90 proteins, 25 and 65 proteins increased and decreased, respectively, by AgNPs mixed with nicotinate and KNO3 compared with control (Table S2).”

-Page 8: the first paragraph of the discussion is a repetition of the introduction, please change it or remove it.

Answer: Thank you very much for your suggestion. Because this sentence is a repetition of the introduction, it has been removed in the discussion section.

-Provide the meaning of PCA and ISR

Answer: As suggested, the meaning of PCA and ISR as well as SAR has been provided in the text in red. They are as follows: “systemic acquired resistance (SAR)”, “induced systemic resistance (ISR)”, and “principal component analysis (PCA)”.

-Page 10 row 275 correct “previous” with “previous”

Answer: I am sorry that we made a mistake. It has been corrected in red.

Reviewer 2 Report

This is a relevant paper about the effect of Silver Nanoparticles when mixed with Organic and Inorganic Chemicals on Wheat Growth.

There are some issues that should be solved before the work is published.

Line 33 Pag 1: It is written “On the otherhand, wheat is vulnerable to…” should be “On the other hand, wheat is vulnerable to…”

Line 40 Pag 2: It is written “The unique antimicrobial activity of silver nanoparticles (AgNPs) provides a strong foundation for the development of biomedical products based on AgNPs, including drug delivery systems, orthopaedic materials/ devices, antiseptic sprays/ catheters, and anticancer agents [5].” Please explain why the antimicrobial activity of silver nanoparticles is important to be used as an anticancer agent. This sentence does not make sense

Line 80 Pag 2: Where it is written “with or without 5 ppm Ag NPs,” should be “with or without 5 ppm AgNPs,”.

Line 172 Pag 5: Figure 1 should be in Materials and Methods and not in results.

Have any tests been carried out regarding the safety and/or toxicity of these nanoparticles after the ingestion of these products?

Author Response

Reviewer 2

This is a relevant paper about the effect of Silver Nanoparticles when mixed with Organic and Inorganic Chemicals on Wheat Growth.

There are some issues that should be solved before the work is published.

Line 33 Pag 1: It is written “On the otherhand, wheat is vulnerable to…” should be “On the other hand, wheat is vulnerable to…”

Answer: Thank you very much for your correction. It has been corrected in red.

Line 40 Pag 2: It is written “The unique antimicrobial activity of silver nanoparticles (AgNPs) provides a strong foundation for the development of biomedical products based on AgNPs, including drug delivery systems, orthopaedic materials/ devices, antiseptic sprays/ catheters, and anticancer agents [5].” Please explain why the antimicrobial activity of silver nanoparticles is important to be used as an anticancer agent. This sentence does not make sense

Answer: Thank you very much for your comment. This sentence has been changed with another reference as follows: “Application of AgNPs increased the plant growth parameters as compared to control, which reduced the percentage of disease incidence as compared to inoculated control R. solani (Das et al., 2021).”

Line 80 Pag 2: Where it is written “with or without 5 ppm Ag NPs,” should be “with or without 5 ppm AgNPs,”.

Answer: I am sorry that we made a mistake. It has been corrected.

Line 172 Pag 5: Figure 1 should be in Materials and Methods and not in results.

Answer: As suggested, Figure 1 has been moved to Materials and Methods.

Have any tests been carried out regarding the safety and/or toxicity of these nanoparticles after the ingestion of these products?

Answer: In this article, the tests related to the safety and/or toxicity of these nanoparticles after the ingestion have not been performed. However, because we performed them in the previous publication (Jhanzab et al., 2019), this information has been cited in the introduction section as follows: “Additionally, morphological study of next generational wheat plants, which were treated with chemo-blended AgNPs, depicted normal growth, and no toxic effects were observed [17]. Therefore, chemo-blended AgNPs may promote plant growth and development through alteration of plant metabolism.”

Reviewer 3 Report

The study of Komatsu and coworkers describes the effect of silver nanoparticles, potassium nitrate and nicotinate on wheat plant growth. Molecular effects are analyzed by mass spectrometry-based proteomics and western blot and biochemical analyses to verify proteome data. Although the experimental part is convincing I do have some questions and remarks regarding the analysis of the data, that I would like to explain in the following.

Methods

In the methods section it is not clear which treatments were used for proteome analysis. Please provide some more details for the LC-MS/MS method. A TopN approach was used – how many precursors per scan cycle were selected for fragmentation? Please add information on the LC gradient length that was used to separate the peptides.

Please add a sentence on parameters for protein identification – how many (unique) peptides were needed for identification of a protein? Could you please give information on the size of the protein database (number of protein entries)?

The authors uploaded their proteome data to the public PRIDE repository. Unfortunately, the reviewer account details were not provided in order to enable the reviewer to have access to the data. Please add the reviewer account details (can be later deleted in a final accepted version of the manuscript).

Line 140 – Which part of the plant was homogenized to analyze the starch content, whole plant, leaves, or roots? Please clarify.

Results/Discussion

Figure 2: Here, error bars for controls are missing. How did the authors calculate significant changes without the standard deviations in the control? I do have some difficulties to understand what „ab“ means with regard to significant changes. „a“ is significantly different from „b“ or „c“ – but it is not quite clear to me, what „ab“ is indicating. In the diagram the values for Leaf weight KNO3 are stated to be significantly different from control. Please provide control data with error bars to make this clear.

Line 200 – With regard to the number of identified proteins the authors refer to Figure S1 and Table S2 for the identification of 5557 proteins. While this is correct for Figure S1, the data in Table S2 show only the proteins with significant changes between the treatments. Please correct and provide a table (excel file) showing all the identified proteins.

Figure S1 - This figure needs improvement, the venn diagram and the accompanying table have poor resolution. What does „search ID:A“ and „search ID:B“ in the table presented in Figure S1 mean? Please rename in „search ID:control“ and „search ID:mixed treatment“ to make the assignment clear. Which data are the basis for the PCA plot shown in Figure S1? Where all proteins used that were identified or just the proteins that remained after the replicate filtering in Perseus software (3 in at least one group as stated in the supplementary table S1)?

Line 203 – After their analysis the authors identified a total of 90 proteins with significantly different amounts between control and mixed treatment. According to the venn diagram depicted in Figure S1 there are about 200 proteins that are exclusively identified search A and more than 400 proteins exclusively identified in search B, respectively. Although these differentially expressed proteins might also be of great interest and might give a much clearer picture of the physiological changes in the wheat plant induced by the mixture of AgNP with KNO3 and nicotinate, the authors did not consider them at all in their analysis. Such so-called „on/off“ proteins are of importance to study the effects of treatments on any biological system. Please include those proteins in your analysis. Furthermore, the analysis of these proteins might support (or not) the conclusions drawn from the existing results by the authors. This also means, that the discussion/conclusions have to be adjusted based on the data revealed by the analysis of these exclusively identified proteins.

Line 204 and Supplementary Table S2 - Among the 90 proteins, 25 showed increased, 65 showed decreased amount in the mixed treatment. To assess the biological relevance of such changes, indications of fold-changes are helpful. The authors provide in table S2 the results of the t-test difference, which is not an useful value for the reader. Induction rates are much easier to understand. Please provide quantitative changes as (log2) fold-changes.

Line 208 – The authors say that proteins with increased amount in the treatment were mainly detected in redox and protein targeting. Actually, the majority of proteins could not be classified – please be more precise and include the analysis of proteins exclusively identified in the control or mixed treatment, respectively.

Figure 3 – In this figure more proteins (3?) from redox than from protein targeting (2?) show higher amounts in the mixed treatment. According to table S2 only two proteins were assigned to redox and two proteins were assigned to protein targeting. Please check.

Line 212 – From the sentence „based on these results, proteins categorized in redox and biotic stress were further confirmed using immunoblot.“ After reading this sentence I would expect, that the same proteins (and maybe additional ones) identified in the proteome analysis were also analyzed by western blot, but the authors use antibodies against different proteins. Why were different proteins chosen to confirm the proteome data?

Line 226 – The header suggests an accumulation of ROS scavening enzymes in mixed treatment, which is not the case according to all the presented data, this only applies to few proteins but not to others involved in ROS detoxification. In my opinion the word „accumulation“ is somewhat misleading here. I would rather use the word „amount“.

Line 233 – (related to comment on line 212) On what basis were the proteins/antibodies selected to confirm the proteome data? In the proteome data catalase and glutathione peroxidase showed increased amount (although unclear how the induction factor is). Why did the authors choose here different proteins? None of them were identified in the proteome analysis.

Line 237 – With reference to Figure 4 the authors say that Peroxiredoxin is decreased in roots. While this is true, there is still a discrepancy between the western blots shown in Fig. 4 and accompanying Figures S2/S3 (see comment below on Figure 4).

Line 238 /Figure S2/S3 – Two bands (~25 and ~60 kDa) are visible on the western blot for ascorbate peroxidase in leafs (Fig S2), the upper one (~ 60 kDa) is marked by the authors. The western blot for ascorbate peroxidase in roots shows only one band (~ 25 kDa). Do leafs express two different ascorbate peroxidases? Why refer the authors to the higher molecular weight band in leafs and not for the smaller one (25 kDa) as in roots?

Figure 4 – The western blot presented in Figure 4 shows clear bands for Peroxiredoxin in leafs as well as in roots. On the accompanying pictures of the whole western blot membranes (Fig. S2, Fig. S3) no bands for Peroxiredoxin are visible.  Are these pictures of the same western blot membranes as in Fig.4? Please check and correct/explain this discrepancy.

Line 250 ff.– I find the annotation of proteins that is used by the authors somewhat hard to follow. They say that they found proteins involved in biotic stress decreased (according to the table S2 these are Uncharacterized proteins and (Endo-)Chitinases); in their chapter 3.4 they talk about PR proteins. What are PR proteins? To someone who is not that familiar with plant physiology the connection between e.g. chitinases and PR proteins is unclear. An explanation (already in the proteome chapter 3.2) would be helpful to follow the author’s experimental approaches. How do PR proteins relate to biotic stress? The authors choose PR1, PR3 and PR5 to be analyzed by western blot. What kind of proteins are these, were they already identified by the proteome analysis? At least in Table S2 no proteins are indicated as PR proteins. Please use consistent protein names and annotations throughout the paper.

Supplementary Figure S5 – for the root samples PR2 is indicated. The figure legend refers to PR1, PR3 and PR5. Please correct.

Line 334 – The sentence is misleading. According to the authors, catalase increased in roots with AgNP and refer to their proteome data. Proteome data were only available for the mixed treatment, but not for AgNP alone. Please correct and clarify.

Line 349 ff. – Some of the proteins involved in ROS detoxification are decreased (glutathione reductase, peroxiredoxin), some are increased (catalase, glutathione peroxidase, ascorbate peroxidase). Why is a repression of some enzymes observed, even if the plants seem to encounter some kind of oxidative stress? I miss here a more in-depth discussion of this observation.

Line 381 – „In this study, the abundance of PR2 and PR3 were suppressed by AgNPs mixed with  nicotinate and KNO3“. Again, the annotation of the proteins is inconsistent. In table S2 no proteins are annotated as PR2 or PR3. Furthermore, according to Fig. 5 only PR1, PR3 and PR5 were analyzed. Where does PR2 come from? Please use consistent annotations for the proteins in the manuscript. The use of different annotations in the tables and in the text makes it hard to follow.

Line 430 ff. Description of contents of table S2 – „List of identified proteins…“ – this is not true; it’s a list of proteins showing significantly different amounts.

My major issue regarding the authors discussion of their proteome data is that they ignore the several hundred proteins that are exclusively found in one of the two tested conditions. If one wants to have a complete picture of physiological changes these proteins must be considered. Additionally, to better evaluate the biological relevance of the detected changes in protein amounts (t-test) fold-changes should be given. Although a fold-change of 1.1 might be statistically significant, it’s biological relevance could be questioned (does such a small change really leads to changes in the physiological status?).

In my opinion, the whole discussion/conclusion should be revised by the authors based on the outcome of the analysis of the so-called on/off proteins. This might give a much clearer picture of the effects of AgNP/KNO3/nicotinate.

Minor

Line 33 – on the other hand, instead of otherhand

Line 48 – AgNP leads to expression of genes related to …, instead of AgNP expressed gene related to…

Line 52 – response to AgNPs, instead of response of AgNPs (which has a totally different meaning)

Line 78 – „AgNPs with 15 nm (US Research Nanomaterials, Houston, TX, USA), nicotinate…“ please change to  „AgNPs (US Research Nanomaterials, Houston, TX, USA), with 15 nm nicotinate…“

Line 79 – Sigma Aldrich Company city and country are missing

Line 110 – Thermo Scientific Company city and country are missing

Line 141 – mortar and pestle

Line 144 – What does „decanted with twice“ mean?

Line 380 – plants instead of palnts

Line 430 ff. – Table S2 does not contain a list of identifie proteins as stated here. It contains a list of proteins with significantly changed proteins. Please correct.

Line 448 – please provide reviewer account details (can be deleted in a later, accepted version of the manuscript).

Author Response

Reviewer 3

The study of Komatsu and coworkers describes the effect of silver nanoparticles, potassium nitrate and nicotinate on wheat plant growth. Molecular effects are analyzed by mass spectrometry-based proteomics and western blot and biochemical analyses to verify proteome data. Although the experimental part is convincing I do have some questions and remarks regarding the analysis of the data, that I would like to explain in the following.

Answer: Thank you very much for your considerable correction.

Methods

In the methods section it is not clear which treatments were used for proteome analysis. Please provide some more details for the LC-MS/MS method.

Answer: As suggested, the part of LC-MS/MS method has been moved from supplemental table 1 to the method section in the text.

A TopN approach was used – how many precursors per scan cycle were selected for fragmentation?

Answer: Because the setting of the system is ”set automatically”, the precursors per cycle could not be selected. The method is as follows: Dynamic exclusion 60 sec, intensity threshold >5.0e3, and MS/MS for 3 sec.  

Please add information on the LC gradient length that was used to separate the peptides.

Answer: As suggested, this explanation has been moved from supplemental table 1 to the method section in the text as follows: “The peptides were loaded onto the LC system (EASY-nLC 1000; Thermo Fisher Scientific, San Jose, CA, USA) equipped with a trap column (Acclaim PepMap 100 C18 LC column, 3 µm, 75 µm ID x 20 mm; Thermo Fisher Scientific) equilibrated with 0.1% formic acid and eluted with a linear acetonitrile gradient (0-35%) in 0.1% formic acid for 120 min at a flow rate of 300 nL/min.”

Please add a sentence on parameters for protein identification – how many (unique) peptides were needed for identification of a protein?

Answer: In the case of Proteome Discoverer, proteins with FDR<0.01 of PSM (peptide spectrum match) are recognized as a protein. Two unique peptides are needed for identification of a protein. This information has been added in supplemental table 1.

Could you please give information on the size of the protein database (number of protein entries)?

Answer: Thank you very much for your suggestion. The information on the size of the protein database has been added in the text in red as follows: “The MS/MS searches were carried out using MASCOT (version 2.6.1, Matrix Science, London, UK) and SEQUEST HT search algorithms against Triticum aestivum (SwissProt TaxID=4565_and_subtaxonomies) (version 2017-07-05) protein database, the size of which is SwissProt=370, TrEBML=145,221, and total=145,591, using Proteome Discoverer (PD) 2.2 (version 2.2.0.388; Thermo Scientific).”

The authors uploaded their proteome data to the public PRIDE repository. Unfortunately, the reviewer account details were not provided in order to enable the reviewer to have access to the data. Please add the reviewer account details (can be later deleted in a final accepted version of the manuscript).

Answer: When this article was submitted to the journal, authors informed the Editor and journal office; however, they did not inform the reviewers. The information provided to the Editor and journal office was as follows: “For MS data, RAW data, peak lists and result files have been deposited in the ProteomeXchange Consortium (Vizcaíno et al., 2013) via the jPOST (Okuda et al., 2017) partner repository under data-set identifiers PXD008949. Access Key is 4048.” We hope that the reviewer can check our MS raw data, peak lists, and results file now.

Line 140 – Which part of the plant was homogenized to analyze the starch content, whole plant, leaves, or roots? Please clarify.

Answer: Leaves were used. This sentence has been changed as follows: “A portion (10 mg) of leaves was homogenized in phosphate-buffered saline with a mortar and pestle.”

Results/Discussion

Figure 2: Here, error bars for controls are missing. How did the authors calculate significant changes without the standard deviations in the control? I do have some difficulties to understand what „ab“ means with regard to significant changes. „a“ is significantly different from „b“ or „c“ – but it is not quite clear to me, what „ab“ is indicating. In the diagram the values for Leaf weight KNO3 are stated to be significantly different from control. Please provide control data with error bars to make this clear.

Answer: I am sorry that we made a mistake when Figure 2 was prepared. Now, control data with error bars have been prepared for Figure 2.

Line 200 – With regard to the number of identified proteins the authors refer to Figure S1 and Table S2 for the identification of 5557 proteins. While this is correct for Figure S1, the data in Table S2 show only the proteins with significant changes between the treatments. Please correct and provide a table (excel file) showing all the identified proteins.

Answer: Thank you very much for your suggestion. New Table S2 has been prepared in Excel, which has all 5557 identified proteins. Previous Table S2 has been changed with a corrected title to table S3.

Figure S1 - This figure needs improvement, the venn diagram and the accompanying table have poor resolution. What does „search ID:A“ and „search ID:B“ in the table presented in Figure S1 mean? Please rename in „search ID:control“ and „search ID:mixed treatment“ to make the assignment clear. Which data are the basis for the PCA plot shown in Figure S1? Where all proteins used that were identified or just the proteins that remained after the replicate filtering in Perseus software (3 in at least one group as stated in the supplementary table S1)?

Answer: We are sorry that Figure S1was not clear and not explained. Because the Venn diagram was automatically generated by the software, an image with high resolution was not prepared; so this graph has been re-written by ourselves. Additionally, the explanation of the PCA picture has been added. Search ID:A and search ID:B mean control and mixture, respectively, and have been renamed. The blue color and orange color show control group and mixture group, respectively. The data used here have been prepared with Excel and this Excel file is provided as the new Table S2.

Line 203 – After their analysis the authors identified a total of 90 proteins with significantly different amounts between control and mixed treatment. According to the venn diagram depicted in Figure S1 there are about 200 proteins that are exclusively identified search A and more than 400 proteins exclusively identified in search B, respectively. Although these differentially expressed proteins might also be of great interest and might give a much clearer picture of the physiological changes in the wheat plant induced by the mixture of AgNP with KNO3 and nicotinate, the authors did not consider them at all in their analysis. Such so-called „on/off“ proteins are of importance to study the effects of treatments on any biological system. Please include those proteins in your analysis. Furthermore, the analysis of these proteins might support (or not) the conclusions drawn from the existing results by the authors. This also means, that the discussion/conclusions have to be adjusted based on the data revealed by the analysis of these exclusively identified proteins.

Answer: The proteins in the Venn diagram are the proteins identified qualitatively by MS analysis. FDR 1% means that one protein in 100 proteins is “False positive”. So, this program is using t-test for comparison quantitatively and qualitatively. After this selection, “on/off proteins” in Figure S1 and Table S2 could not select the significant proteins. As a result, we could not discuss or make any conclusions about these proteins.

Line 204 and Supplementary Table S2 - Among the 90 proteins, 25 showed increased, 65 showed decreased amount in the mixed treatment. To assess the biological relevance of such changes, indications of fold-changes are helpful. The authors provide in table S2 the results of the t-test difference, which is not an useful value for the reader. Induction rates are much easier to understand. Please provide quantitative changes as (log2) fold-changes.

Answer: Thank you very much for your comments. The word “difference” in new Table S3 means “fold change”, not “LOG(P-Value)” of t-test. To clarify, this word has been changed to the word “fold change” in the new Table S3.  

Line 208 – The authors say that proteins with increased amount in the treatment were mainly detected in redox and protein targeting. Actually, the majority of proteins could not be classified – please be more precise and include the analysis of proteins exclusively identified in the control or mixed treatment, respectively.

Answer: Thank you very much for your suggestion. The proteins in the Venn diagram are the proteins identified qualitatively by MS analysis. Using t-test for comparison quantitatively and qualitatively, the proteins were selected. After this selection, “on/off proteins” in Figure S1 and Table S2 could not select the significant proteins. So, we could not use them as potential proteins.

Figure 3 – In this figure more proteins (3?) from redox than from protein targeting (2?) show higher amounts in the mixed treatment. According to table S2 only two proteins were assigned to redox and two proteins were assigned to protein targeting. Please check.

Answer: We are sorry we made a mistake. Based on the new Table S3, Figure 3 has been re-prepared and the result section has been corrected based on the revised Figure 3. The corrected parts are as follows: “The identified proteins were functionally categorized using MapMan bin codes (Table S3, Figure 3). The differentially changed proteins were mainly detected in redox and biotic stress in the functional category (Figure 3).”

Line 212 – From the sentence „based on these results, proteins categorized in redox and biotic stress were further confirmed using immunoblot.“ After reading this sentence I would expect, that the same proteins (and maybe additional ones) identified in the proteome analysis were also analyzed by western blot, but the authors use antibodies against different proteins. Why were different proteins chosen to confirm the proteome data?

Answer: Thank you very much for your critical question. Some of the proteins were identified as changed proteins categorized in redox and biotic stress by proteomic analysis. So, other major proteins categorized in redox and biotic stress have been confirmed because of the relationship between these two functional categories and the effect of the AgNPs/ nicotinate/ KNO3 mixture. The reasons have been added in the results sections “3.3” and “3.4” in red.  

Line 226 – The header suggests an accumulation of ROS scavening enzymes in mixed treatment, which is not the case according to all the presented data, this only applies to few proteins but not to others involved in ROS detoxification. In my opinion the word „accumulation“ is somewhat misleading here. I would rather use the word „amount“.

Answer: The word “accumulation” has been corrected to “amount” for ROS scavenging enzymes as suggested.

Line 233 – (related to comment on line 212) On what basis were the proteins/antibodies selected to confirm the proteome data? In the proteome data catalase and glutathione peroxidase showed increased amount (although unclear how the induction factor is). Why did the authors choose here different proteins? None of them were identified in the proteome analysis.

Answer: Because glutathione peroxidase and catalase were identified as increased proteins by proteomic analysis, other proteins, which were ascorbate peroxidase, glutathione reductase, and peroxiredoxin were analyzed by immunoblot analysis. This reason has been added in the results section “3.4” in red.

Line 237 – With reference to Figure 4 the authors say that Peroxiredoxin is decreased in roots. While this is true, there is still a discrepancy between the western blots shown in Fig. 4 and accompanying Figures S2/S3 (see comment below on Figure 4).

 Answer: We are sorry about this problem. We have performed the immunoblot analysis with more than 5 times for each. Now, Figures S2 and S3 have been changed with the picture used in Figure 4.

Line 238 /Figure S2/S3 – Two bands (~25 and ~60 kDa) are visible on the western blot for ascorbate peroxidase in leafs (Fig S2), the upper one (~ 60 kDa) is marked by the authors. The western blot for ascorbate peroxidase in roots shows only one band (~ 25 kDa). Do leafs express two different ascorbate peroxidases? Why refer the authors to the higher molecular weight band in leafs and not for the smaller one (25 kDa) as in roots?

 Answer: The anti-ascorbate peroxidase antibody used in this research could detect cytosolic ascorbate peroxidase, which is 25 kDa, and chloroplast/mitochondria ascorbate peroxidase, which is 60 kDa (Komatsu et al., 2011). Although the band of chloroplast/mitochondria ascorbate peroxidase was selected, the band of cytosolic ascorbate peroxidase was re-analyzed and changed in Figure 4. Based on this change, result section “3.3” and Figure S2 have been corrected in red.

Figure 4 – The western blot presented in Figure 4 shows clear bands for Peroxiredoxin in leafs as well as in roots. On the accompanying pictures of the whole western blot membranes (Fig. S2, Fig. S3) no bands for Peroxiredoxin are visible.  Are these pictures of the same western blot membranes as in Fig.4? Please check and correct/explain this discrepancy.

 Answer: We are sorry about this problem. We have performed the immunoblot analysis more than 5 times for each. Now, Figures S2 and S3 have been changed with the picture used in Figure 4.

Line 250 ff.– I find the annotation of proteins that is used by the authors somewhat hard to follow. They say that they found proteins involved in biotic stress decreased (according to the table S2 these are Uncharacterized proteins and (Endo-)Chitinases); in their chapter 3.4 they talk about PR proteins. What are PR proteins? To someone who is not that familiar with plant physiology the connection between e.g. chitinases and PR proteins is unclear. An explanation (already in the proteome chapter 3.2) would be helpful to follow the author’s experimental approaches. How do PR proteins relate to biotic stress? The authors choose PR1, PR3 and PR5 to be analyzed by western blot. What kind of proteins are these, were they already identified by the proteome analysis? At least in Table S2 no proteins are indicated as PR proteins. Please use consistent protein names and annotations throughout the paper.

 Answer: We are sorry about this problem. Chitinase named PR3 and thaumatin named as PR5 were identified as decreased proteins by proteomic analysis. An explanation has been added in the section “3.4” and “4.3”.

Supplementary Figure S5 – for the root samples PR2 is indicated. The figure legend refers to PR1, PR3 and PR5. Please correct.

 Answer: We are sorry for this mistake. They have been corrected.

Line 334 – The sentence is misleading. According to the authors, catalase increased in roots with AgNP and refer to their proteome data. Proteome data were only available for the mixed treatment, but not for AgNP alone. Please correct and clarify.

 Answer: We are sorry for this problem, again. This mistake has been corrected in discussion section “4.2” in the text in red.

Line 349 ff. – Some of the proteins involved in ROS detoxification are decreased (glutathione reductase, peroxiredoxin), some are increased (catalase, glutathione peroxidase, ascorbate peroxidase). Why is a repression of some enzymes observed, even if the plants seem to encounter some kind of oxidative stress? I miss here a more in-depth discussion of this observation.

Answer: We are sorry this problem. The discussion section “4.2” has been extended as suggested. The corrected parts have been marked in the text in red.

Line 381 – „In this study, the abundance of PR2 and PR3 were suppressed by AgNPs mixed with  nicotinate and KNO3“. Again, the annotation of the proteins is inconsistent. In table S2 no proteins are annotated as PR2 or PR3. Furthermore, according to Fig. 5 only PR1, PR3 and PR5 were analyzed. Where does PR2 come from? Please use consistent annotations for the proteins in the manuscript. The use of different annotations in the tables and in the text makes it hard to follow.

Answer: We are sorry for this mistake. The sentence has been corrected as follows: “In this study, the abundance of chitinase named PR3 and thaumatin named as PR5 were suppressed by AgNPs mixed with nicotinate and KNO3 (Table S3, Figure 5).” Furthermore, because the name of proteins in the new Table S3 was from the protein database, we could not change it. So, the names of PR proteins are explained in the text.

Line 430 ff. Description of contents of table S2 – „List of identified proteins…“ – this is not true; it’s a list of proteins showing significantly different amounts.

My major issue regarding the authors discussion of their proteome data is that they ignore the several hundred proteins that are exclusively found in one of the two tested conditions. If one wants to have a complete picture of physiological changes these proteins must be considered.

Answer: Thank you very much for your correction. The title of Table S3, which is previous Table S2, has been corrected as suggested. The 5557 proteins in the Venn diagram are the proteins identified qualitatively by MS analysis. Using t-test for comparison quantitatively and qualitatively, they were selected. After this selection, the several hundred proteins could not be selected as the significant proteins. So, we could not use them as potential proteins. Based on confirmation experiments, discussion section has been corrected in red.

Additionally, to better evaluate the biological relevance of the detected changes in protein amounts (t-test) fold-changes should be given. Although a fold-change of 1.1 might be statistically significant, it’s biological relevance could be questioned (does such a small change really leads to changes in the physiological status?).

Answer: As suggested, P-Value by t-test has been added in the revised Table S3. Although fold-change is small, we are using these proteins because t-test informs that they are statistically significant. Because the reviewer’s opinion is acceptable, we are additionally doing the confirmation experiments.

In my opinion, the whole discussion/conclusion should be revised by the authors based on the outcome of the analysis of the so-called on/off proteins. This might give a much clearer picture of the effects of AgNP/KNO3/nicotinate.

Answer: Thank you very much for your suggestion. In this article, “on/off proteins” in Figure S1 and Table S2 could not select the significant proteins by t-test in the end. So, we could not discuss or make any conclusion about these proteins. However, because the comments from the reviewer are very important, we will consider this issue in another article in the future.

Minor

Line 33 – on the other hand, instead of otherhand

Line 48 – AgNP leads to expression of genes related to …, instead of AgNP expressed gene related to…

Line 52 – response to AgNPs, instead of response of AgNPs (which has a totally different meaning)

Line 78 – „AgNPs with 15 nm (US Research Nanomaterials, Houston, TX, USA), nicotinate…“ please change to  „AgNPs (US Research Nanomaterials, Houston, TX, USA), with 15 nm nicotinate…“

Answer: Thank you very much for your suggestions and I am sorry we made mistakes. All parts have been corrected based on your suggestions.

Line 79 – Sigma Aldrich Company city and country are missing

Answer: Thank you very much for your suggestion. About Sigma Aldrich company, this is the second time, it was mentioned; so, city and country had been deleted.

Line 110 – Thermo Scientific Company city and country are missing

Answer: Thank you very much for your kind correction and it has been added as follows: “Thermo Scientific, San Jose, CA, USA”

Line 141 – mortar and pestle

Line 144 – What does „decanted with twice“ mean?

Line 380 – plants instead of palnts

We are sorry we made mistakes. All parts have been corrected based on these suggestions.

Line 430 ff. – Table S2 does not contain a list of identifie proteins as stated here. It contains a list of proteins with significantly changed proteins. Please correct.

Answer: As suggested, new Table S2 has been prepared and revised Table S3, which is previous Table S2.

Line 448 – please provide reviewer account details (can be deleted in a later, accepted version of the manuscript).

Answer: When this article was submitted to the journal, authors informed the Editor and journal office; however, they did not inform the reviewers. The information provided to the Editor and journal office was as follows: “For MS data, RAW data, peak lists and result files have been deposited in the ProteomeXchange Consortium (Vizcaíno et al., 2013) via the jPOST (Okuda et al., 2017) partner repository under data-set identifiers PXD008949. Access Key is 4048.” We hope that the reviewer can check our MS raw data, peak lists, and results file now.

Round 2

Reviewer 2 Report

The authors have answered the questions and changed as asked.

Author Response

Reviewer 2

The authors have answered the questions and changed as asked.

Answer: Thank you very much for your correction in the first version.

Reviewer 3 Report

Thank you for answering my comments and suggestions in detail.

I still have a question about table S3: As suggested the authors changed the induction ratios and indicate now fold changes in the first column in this table.  However, fold changes can only have positive values; values >1 mean induction regarding to the control; fold changes <1 mean repression regarding to the control. In this table also negative values for fold changes are shown. Negative values are only possible when you use "log2 fold changes". Please check, which values ("fold change" or "log2 fold change") are shown here and rename the column header accordingly.

Author Response

Reviewer 3

I still have a question about table S3: As suggested the authors changed the induction ratios and indicate now fold changes in the first column in this table.  However, fold changes can only have positive values; values >1 mean induction regarding to the control; fold changes <1 mean repression regarding to the control. In this table also negative values for fold changes are shown. Negative values are only possible when you use "log2 fold changes". Please check, which values ("fold change" or "log2 fold change") are shown here and rename the column header accordingly.

Answer: We are sorry we made a mistake. It is “log2 (fold change)”. It has been corrected in Table S3.